# Conditioned respiratory threat in the subdivisions of the human periaqueductal gray

**Olivia K Faull[1,2]\*, Mark Jenkinson[1], Martyn Ezra[1,2], Kyle TS Pattinson[1,2]\***

[1]Oxford Centre for Functional MRI of the Brain, University of Oxford, Oxford, United Kingdom; [2]Nuffield Division of Anesthetics, Nuffield Department of Clinical Neurosciences, University of Oxford, Oxford, United Kingdom

**Abstract** The sensation of breathlessness is the most threatening symptom of respiratory disease. The different subdivisions of the midbrain periaqueductal gray (PAG) are intricately (and differentially) involved in integrating behavioural responses to threat in animals, while the PAG has previously only been considered as a single entity in human research. Here we investigate how these individual PAG columns are differently involved with respiratory threat. Eighteen healthy subjects were conditioned to associate shapes with certain or uncertain impending respiratory load, and scanned the following day during anticipation and application of inspiratory loading using 7 T functional MRI. We showed activity in the ventrolateral PAG (vlPAG) during anticipation of resistive loading, with activity in the lateral PAG (lPAG) during resistive loading, revealing spatially and temporally distinct functions within this structure. We propose that lPAG is involved with sensorimotor responses to breathlessness, while the vlPAG operates within the threat perception network for impending breathlessness.

**\*For correspondence:** olivia.faull@ndcn.ox.ac.uk (OKF); kyle.pattinson@ndcn.ox.ac.uk (KTP)

**Competing interests:** The authors declare that no competing interests exist.

## Introduction

Continued respiratory function is crucial for sustaining life, and perceived threat to respiration can induce an integrated stress reaction and crippling anxiety. A potentially pivotal nucleus within the breathlessness perception pathway is the midbrain periaqueductal gray (PAG). The PAG has been implicated in many basic survival behaviours including cardiovascular, motor and pain responses (*De Oca et al., 1998*; *Mobbs et al., 2007*; *Pereira et al., 2010*; *Tracey et al., 2002*; *Benarroch, 2012*; *Paterson, 2014*), and is well situated to play a role in the integrative response to breathlessness, with major cortical inputs from areas involved with emotional regulation such as medial prefrontal, insula, anterior cingulate cortices and amygdala (*Beitz, 1982*; *Gabbott et al., 2005*; *Rizvi et al., 1991*), and descending connections to the respiratory nuclei of the medulla (*Huang et al., 2000*; *Sessle et al., 1981*; *Hayward, 2004*). These medullary nuclei projections include those to the ventrolateral medulla for the switch between inspiration and expiration (*Subramanian et al., 2008*; *Subramanian, 2013*), and the nucleus retroambiguus for pharyngeal, laryngeal, thoracic and abdominal pressure control (*Holstege, 1989*).

The PAG is subdivided into four columns either side of the aqueduct (ventrolateral (vlPAG), lateral (lPAG), dorsolateral (dlPAG) and dorsomedial (dmPAG)), each with distinct functions for the control of respiration (*Subramanian et al., 2008*; *Subramanian, 2013*). These columns are proposed to act within active coping strategies for escapable threat (lPAG and dlPAG) associated with fight and flight responses, or passive coping strategies to inescapable threat (vlPAG) often associated with freezing behaviours (*Keay and Bandler, 2001*; *Bandler and Shipley, 1994*; *Bandler et al., 2000*).

**eLife digest** Many people find feeling breathless one of the most upsetting symptoms of respiratory diseases, and breathlessness often causes anxiety that makes the condition seem more threatening than it is. Studies in animals suggest that a small cluster of neurons called the periaqueductal gray is important for responding to threats. This cluster, located at the top of the brainstem, is divided into parallel columns running from top to bottom. In animals, these columns are known to have distinct roles, but human research has tended to consider the periaqueductal gray as a single, uniform entity.

Faull et al. wanted to find out whether different columns of the human periaqueductal gray have distinct roles in the perception of respiratory threat. During the study, participants breathed through a tube while watching shapes appear on a screen. This tube could be altered to make breathing more or less difficult – much like breathing through a narrow drinking straw.

A conditioning session was first conducted so that participants learned that certain shapes on the screen signalled that their breathing was about to become difficult, while other shapes signalled normal breathing. A second session was then conducted in a brain scanner, using a technique called functional magnetic resonance imaging. This allowed Faull et al. to compare brain activity during the anticipation of difficult breathing with the brain activity during the breathing challenge itself.

The results show that the column at the front of the periaqueductal gray (the ventrolateral column) was more active when participants saw the shape that signaled upcoming breathing difficulty. In contrast, difficult breathing was associated with activity in the lateral column (at the side of the periaqueductal gray).

Thus, the different columns of the human periaqueductal gray have different roles in the response to respiratory threat. Future studies could investigate how these columns interact with each other and with other brain regions. Such understanding is important for a range of conditions that may be influenced by the activity of the periaqueductal gray, including disruptions in bladder control, hypertension, chronic pain, and asthma.

However, the roles of the different columns in human respiratory threat perception are yet to be investigated.

Clinical populations such as those with chronic obstructive lung disease (COPD), asthma, heart failure, cancer and panic disorder suffer from debilitating breathlessness, that contributes to a downward spiral of reduced physical activity, physical deconditioning and worsening breathlessness (*Hayen et al., 2013a*). Therefore, a better understanding of its neural basis has the potential to lead to new treatments with wide ranging impact (*Herigstad et al., 2011*). Importantly, conditioned anticipation of environmental cues associated with breathlessness is integral to its threat detection and designated response. Anticipation of threatening sensations relies on cues from the environment. Conditioning is a process of learning an association between two unrelated stimuli, such that a previously neutral stimulus (conditioned stimulus, CS) may evoke anxiety due to learned associations with an aversive stimulus (unconditioned stimulus, US) (*Pavlov et al., 2003*). Descending modulatory systems during anticipation of a stimulus have even been shown to modulate the response to the stimulus itself, such as those demonstrated with pain (*Porro et al., 2002*; *Price et al., 1999*; *Wager et al., 2004*). Therefore, each individual PAG column may play differential and important roles in both the anticipation and response to a threatening respiratory stimulus, and thus are potentially pivotal in our understanding and treatment of the neural basis of breathlessness. Interestingly, PAG activity has been identified in a recent paper investigating brain responses to breathlessness-related word cues in patients with COPD, although without sufficient resolution to differentiate activity within specific columns (*Herigstad et al., 2015*).

Our understanding of anticipation of conditioned threat has been substantially enhanced by modern neuroimaging techniques, and despite differences in conditioning paradigms, a consistent network of brain areas has been identified, including the amygdala, insula, and anterior cingulate cortex (*Sehlmeyer et al., 2009*). However, despite the proposed integral role of the PAG in threat perception, the ability to scrutinise contributions of smaller nuclei is often limited in neuroimaging by

resolution and statistical power, and thus key structures such as the PAG have not yet been investigated in humans.

The aim of this study was to investigate the roles of the individual PAG columns during both the perception of breathlessness and its anticipation. We used an aversive delay-conditioning paradigm to associate neutral shapes with upcoming resistive loaded breathing. To investigate if uncertainty of an aversive breathing stimulus altered the threat response (*Rhudy and Meagher, 2000*; *Ploghaus et al., 2003*), we used three separate anticipation cues with a CS-US contingency pairing of 100%, 50% and 0%. In accordance with animal models, we hypothesised that the vlPAG would be active during anticipation of resistance, as threat is detected and passive coping strategies are employed to manage an upcoming inescapable stressor. Conversely, we hypothesised activity in the lPAG during inspiratory resistance with slowed, deep breathing, corresponding with results in animals (*Holstege, 1989*; *Keay and Bandler, 2001*) and our previous work in breath holds (*Faull et al., 2015*).

## Results

### Behavioural scores

Mean anxiety and intensity scores for conditioned responses to the respiratory tasks are given in *Table 1*. Anxiety scores were significantly higher for the certain anticipation cue compared to the uncertain cue, and subsequent resistance was rated at a greater intensity following the certain cue.

### Physiology

Group average heart rate ( ± SD) during the brainstem BOLD scanning was 68 ( ± 9) beats per minute. Ventilatory variables during each of the respiratory conditions are given in *Table 2*. Certain anticipation of resistance was associated with a greater decrease in $P_{ET}CO_2$ and increase in $P_{ET}O_2$ and respiratory volume per unit of time (RVT) than uncertain, indicating preparatory increases in respiration with more effective conditioning.

### Periaqueductal gray fMRI analysis

The results of the targeted PAG subdivision analyses (in which certain anticipation of resistive loading was contrasted with anticipation of no loading) revealed significant increased BOLD activity in the vlPAG, and decreased BOLD in the lPAG during inspiratory resistance (*Figure 1*). A further analysis of the whole PAG showed that these activations were isolated to the vlPAG and lPAG in these conditions, although certain anticipation of resistance was now analysed against baseline for adequate statistical power (*Figure 2*). Furthermore, activity in the lPAG during certain anticipation of resistance was found to scale with intensity ratings across subjects (*Figure 3*). No areas of the PAG or cortex significantly scaled with intensity or anxiety ratings during inspiratory loading, possibly due to insufficient statistical power necessary to observe these scaled activations across subjects during the noisy stimulus of inspiratory loading.

When comparing uncertain and certain anticipation of breathlessness, no significant difference was found in the PAG between the two conditions, possibly due to insufficient statistical power to detect a difference. However, during uncertain anticipation of resistance, subthreshold PAG activity (p=0.11) was identified in the same area of the right vlPAG as the significant cluster found with certain anticipation of resistance (*Figure 4*). Activity in neither the vlPAG, nor the lPAG scaled with anxiety across subjects.

**Table 1.** Mean ( ± SD) anxiety and intensity ratings to the conditioned respiratory tasks.

|  | No impending resistance | Uncertain impending resistance | Certain impending resistance |
|---|---|---|---|
| Anxiety (%) | 4.3 (5.1) | 36.7 (22.3)* | 48 (26.7)** |
| Intensity (%) | 4.7 (3.1) | 55.5 (20.9)* | 62.9 (21.5)** |

*Significantly (p<0.05) different from 'no impending resistance' condition;
**Significantly (p<0.05) different from 'no impending resistance' and 'uncertain impending resistance'.

**Table 2.** Mean ( ± SD) physiological variables across conditioned respiratory tasks.

| | Anticipation | | | Resistance | |
|---|---|---|---|---|---|
| | No impending resistance | Uncertain impending resistance | Certain impending resistance | Average | Peak |
| Pressure (cmH$_2$O) | -0.14 (0.11) | -0.17 (0.12) | -0.18 (0.24) | -5.80 (3.64)* | -14.67 (8.28)* |
| P$_{ET}$CO$_2$ (%) | 4.41 (0.71) | 4.41 (0.67) | 4.32 (0.68)* | 4.46 (0.67) | 4.62 (0.66)* |
| P$_{ET}$O$_2$ (%) | 18.1 (1.0) | 18.1 (1.0) | 18.3 (1.1)* | 18.5 (1.0)* | 18.9 (1.0)* |
| Respiratory rate (min$^{-1}$) | 12.8 (3.7) | 12.5 (3.8) | 12.4 (3.6) | 11.2 (4.6) | 13.8 (5.9) |
| RVT increase (%) | -4.4 (7.4) | 7.8 (19.6)* | 11.0 (23.0)* | -16.1 (21.6)* | 16.6 (28.5)* |

*Significantly (p<0.05) different from 'no impending resistance' condition.

Abbreviations: Pressure, average mouth pressure across all ventilatory cycles; P$_{ET}$CO$_2$, pressure of end-tidal carbon dioxide; P$_{ET}$O$_2$, pressure of end-tidal oxygen; RVT, respiratory volume per unit time.

## Cortical and subcortical respiratory results

All respiratory tasks: We observed significant BOLD signal increases bilaterally in the motor cortex, supplementary motor cortex, primary sensory cortex, middle and posterior cingulate cortices, operculum, medulla and middle insular cortex, and decreased BOLD signal in the bilateral hippocampus and IX cerebellar lobe, for both certain and uncertain anticipation against baseline, and during inspiratory resistance (*Figure 5*).

### Anticipation

Both certain and uncertain anticipation additionally correlated with bilateral deactivations in the posterior insula. No significant cortical or subcortical differences were seen between certain and uncertain anticipation.

### Resistive loading

Inspiratory resistance also correlated with activations in the bilateral putamen, caudate, ventral posterior lateral nucleus (thalamus) and subthalamic nucleus, and deactivations in the bilateral amygdala, lPAG and posterior nuclei of the thalamus.

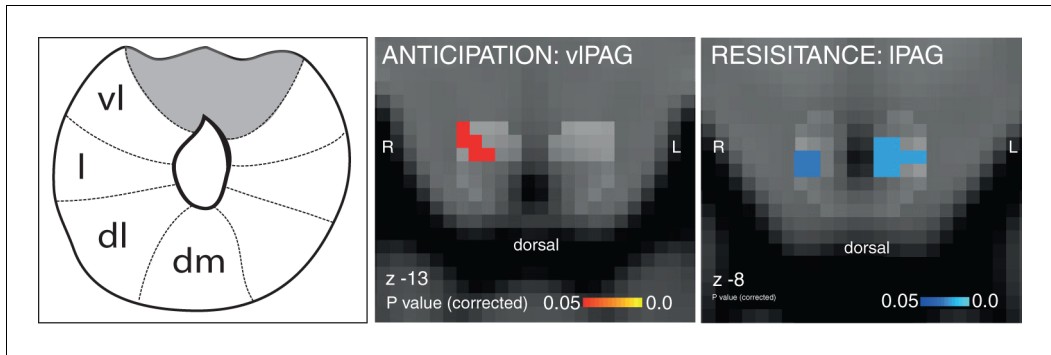

**Figure 1.** Targeted PAG columnar analysis. Left: Schematic representation of the columns of the midbrain periaqueductal gray (PAG), which almost surrounds the aqueduct. Middle: Ventrolateral PAG (vlPAG) activation during anticipation of resistance contrasted with anticipation of no resistance. Right: Lateral PAG (lPAG) deactivation during inspiratory resistance. Statistics are small-volume-corrected for multiple comparisons using highlighted PAG column masks, adapted from *Ezra et al. (2015)*, and the images consist of a colour-rendered statistical map superimposed on a standard (MNI 1 mm$^3$) brain. Line drawing originally published in *Ezra et al., 2015*.

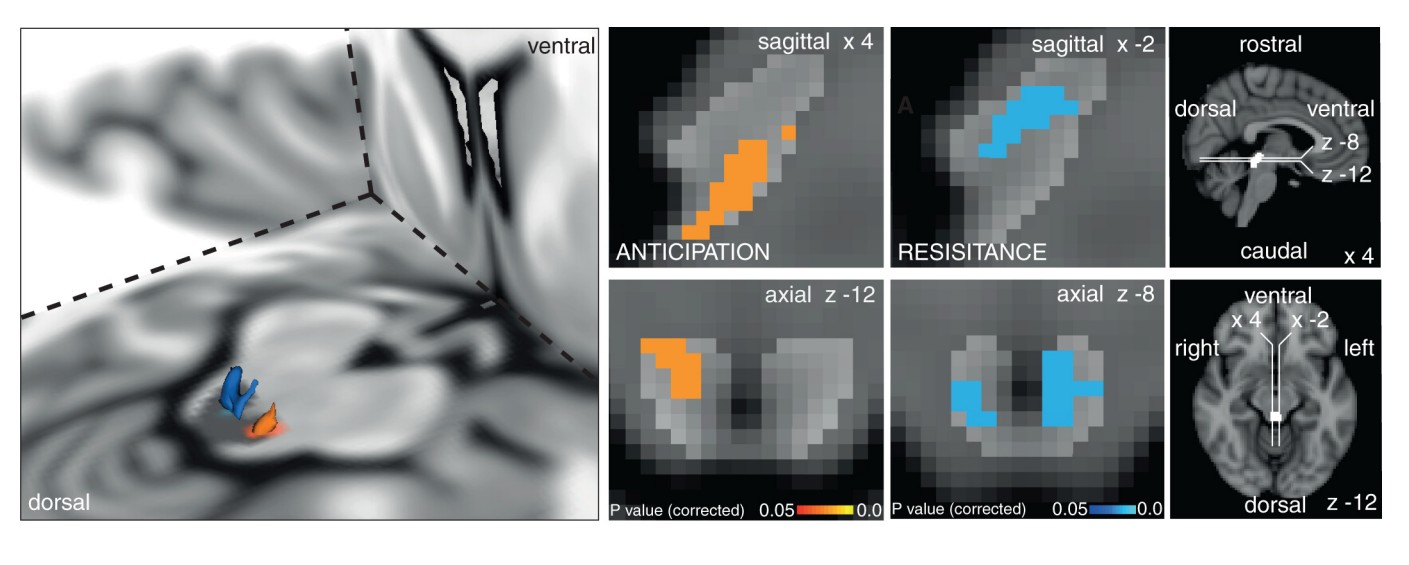

**Figure 2.** Periaqueductal gray (PAG) analysis. Left: 3D representation of the PAG activations on the right. Top row sagittal view, bottom row axial view of activation in the vlPAG during anticipation of certain resistance (against baseline: p=0.021) and deactivation during inspiratory resistance in bilateral lPAG (p=0.007). The key on the right shows location of PAG mask and orientation of displayed slices. Statistics are small-volume-corrected for multiple comparisons using highlighted PAG mask, and the images consist of a colour-rendered statistical map superimposed on a standard (MNI 1 mm$^3$) brain.

The hypercapnia challenges and the resultant $CO_2$ regressor produced strong BOLD signal increases throughout the grey matter of the brain. Furthermore, increases in BOLD signal correlating to the $CO_2$ regressor were observed within the PAG, localised to the grey matter and excluding the aqueduct (*Figure 6*).

## Finger opposition task

Finger opposition resulted in consistant significant signal increases in both the brainstem and motor cortex with previous research (*Faull et al., 2015*; *Lee et al., 1999*; *Pattinson et al., 2009*) including bilateral activation in the motor cortex (more extensive activation in the contralateral left motor cortex), supplementary motor cortex, middle cingulate and paracingulate cortices, primary sensory cortex, anterior insula cortex, operculum, caudate nucleus and putamen (*Figure 5*). Bilateral signal increases were seen in the thalamic VPL nuclei, as well as the left thalamic VPM nucleus. In addition, activations were observed in the left subthalamic and red nuclei, right (ipsilateral) cuneate nucleus of the medulla (*Figure 7*), and bilateral cerebellum (VI and VIIIa lobules).

## Discussion

## Main findings

In this study we identified differential activity in the lateral and ventrolateral columns of the PAG relating to different aspects of the aversive stimulus of resistive inspiratory loading. We observed bilateral decreased BOLD activity in the lPAG during resistive inspiratory loading, and during cued anticipation activity in this area correlated with behavioural ratings of breathlessness intensity. Conversely, positive BOLD activity in the right vlPAG was identified during the cued anticipation of certain impending resistance, while uncertain anticipation activity remained subthreshold. Anxiety ratings, intensity scores and the ventilatory response were lower in the uncertain vs. certain condition, indicating a reduction in the conditioned threat response to a 50% (uncertain) predictive cue, compared to the 100% (certain) predictive cue.

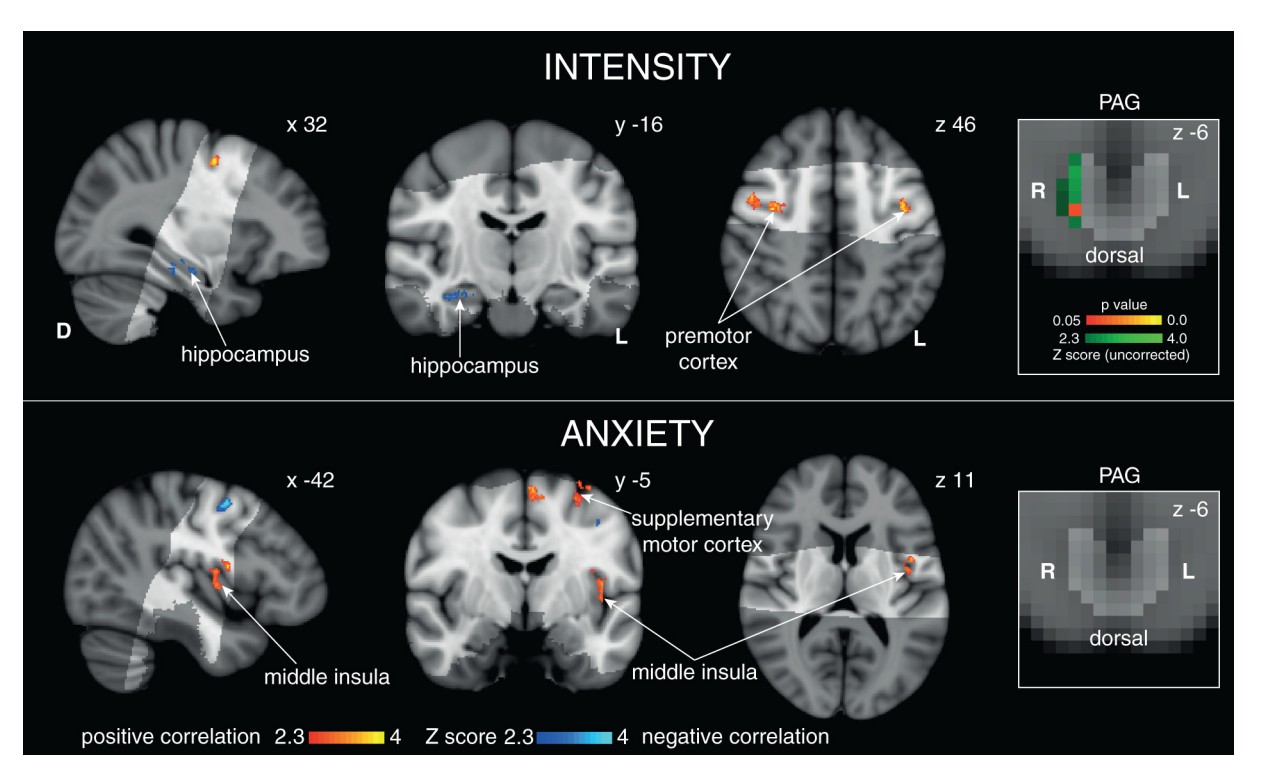

**Figure 3.** Scaled BOLD activity during 100% certain anticipation with intensity and anxiety. Right: Positive correlation in the lPAG with intensity ratings (green uncorrected Z score, red/yellow TFCE-corrected for lPAG activity, PAG displayed in light grey) but not anxiety. Top: Cortical correlations with average intensity score. Bottom: Cortical correlations with anxiety score for certain anticipation. Images consist of a colour-rendered statistical map superimposed on a standard (MNI 1 mm$^3$) brain.

## PAG and threat

Significantly, recent work using diffusion tractography has revealed consistent columnar structure to animal models within the human PAG (*Ezra et al., 2015*). During response to threat, functional organisation of these animal PAG columns has been hypothesised to consist of active and passive coping strategies (*Bandler et al., 2000*; *Hayen et al., 2013a*; *Herigstad et al., 2011*). The lPAG and dlPAG are thought to employ active coping strategies for escapable stressors, consistent with the tachypnea observed in animals on stimulation of these columns (*Holstege, 1989*), while the vlPAG employs passive coping strategies for inescapable stressors (vlPAG) such as that seen with a range of physical stimuli (*Bandler et al., 2000*). In the current investigation of the threat response to breathlessness, the aversive resisted breathing stimulus was an upcoming inescapable stressor, activating the vlPAG, while during the active stimulus response we observed lPAG activity. These results are the first in humans to adhere to the current models of distinctive threat perception of the animal PAG columns, although the cytoarchitecture and autonomic functions produced within the rostro-caudal axes of these columns is in humans not yet known. One recent study by Satpute and colleagues used 7 Ttesla functional MRI to identify highly localised activity in areas of the PAG along the rostro-caudal axis during exposure to aversive images in humans [*Satpute et al., 2013*] however this study was based upon pre-defined divisions within the PAG that neither adhered to its known columnar structure, nor considered the characteristic functions of these columns within threat perception. Therefore, while there is much work to be done to accurately map the cytoarchitecture and functional localisation of autonomic functions within the human PAG columns, we will now discuss each of the activated columns in the current study, as a starting point towards understanding their potential role in the specific threat response to an aversive breathing stimulus as a model of breathlessness.

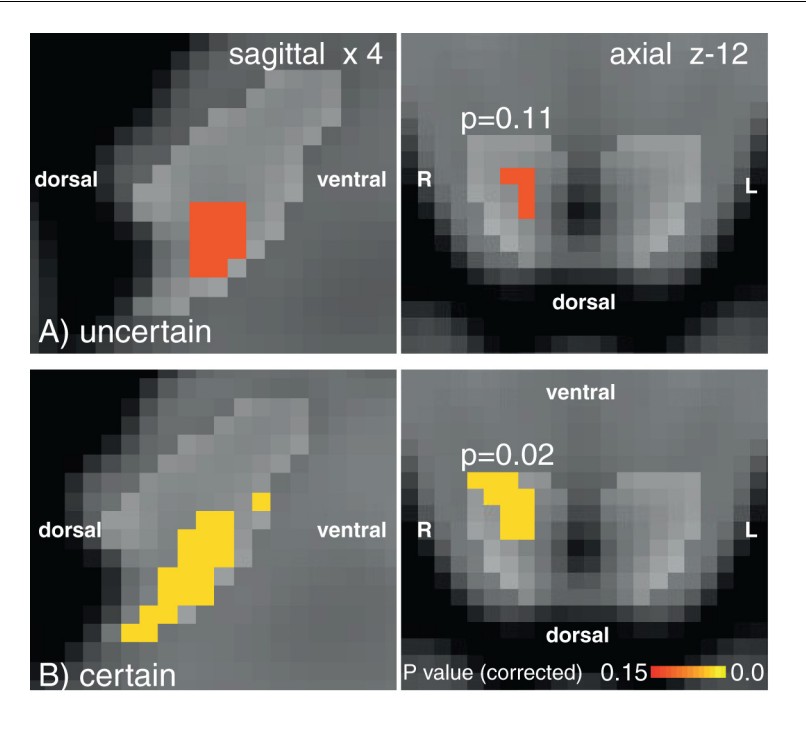

**Figure 4.** vlPAG activation with 100% certainty of resistance. vlPAG activations during uncertain (**A**) and certain (**B**) anticipation of impending breathlessness. Uncertain anticipation produces subthreshold vlPAG activation in a consistent area to the certain condition. PAG mask displayed by light grey region. Images consist of a colour-rendered statistical map superimposed on a standard (MNI 1 mm³) brain. Orientations marked on the image.

## lPAG in conditioned breathlessness

The decrease in BOLD signal in the lPAG during an inspiratory resistance found in this study is consistent with previous PAG findings. Prior work by our group identified decreased BOLD signal in the human lPAG during the respiratory challenge of breath holds (*Faull et al., 2015*), and animal studies have proposed the lPAG may play a role in respiratory behaviours such as prolonged inspirations and expirations (*Holstege, 1989*). Thus, it is possible that the lPAG is an integral nucleus within the somatomotor pathways of respiratory control in the active response to threat, and anatomical evidence exists to support this hypothesis (*Ezra et al., 2015*). The lPAG has been reported to receive somatotopically organised spinal sensory afferents (*Bandler et al., 2000*; *Craven, 2011*), which could provide sensory information from the chest, and it propagates direct efferent connections to the midline medulla (*Cowie and Holstege, 1992*) for possible descending respiratory motor commands. Diffusion tractography in humans demonstrates preferential connectivity between somato-motor regions, such as between primary sensory and motor cortices and the lPAG, compared to the vlPAG (*Ezra et al., 2015*). Our findings of activity in the lPAG whilst producing elevated inspiratory pressure supports the idea that this column of the PAG is involved in altered respiratory work, although whether this is in a motor or sensory capacity (or both) is currently unknown.

Interestingly, activity in the lPAG during anticipation was found to scale with perceived stimulus intensity across subjects. Anticipation of a stimulus allows system preparation and response selection, and activity in the lPAG that scales with the perceived intensity of the forthcoming stimulus indicates a possible top-down control during preparation for the threat of inspiratory resistance. The cortical structures that scaled alongside the lPAG with perceived intensity included the premotor cortex and hippocampus, which may indicate increased motor preparatory activity (*Grafton et al., 1998*; *Rizzolatti et al., 2002*) and greater working memory of the stimulus between the hippocampus and prefrontal cortex (*Laroche et al., 2000*). Conversely, lPAG activity during anticipation did not correlate with anxiety scores. This suggests that lPAG activity is less likely to be involved in the

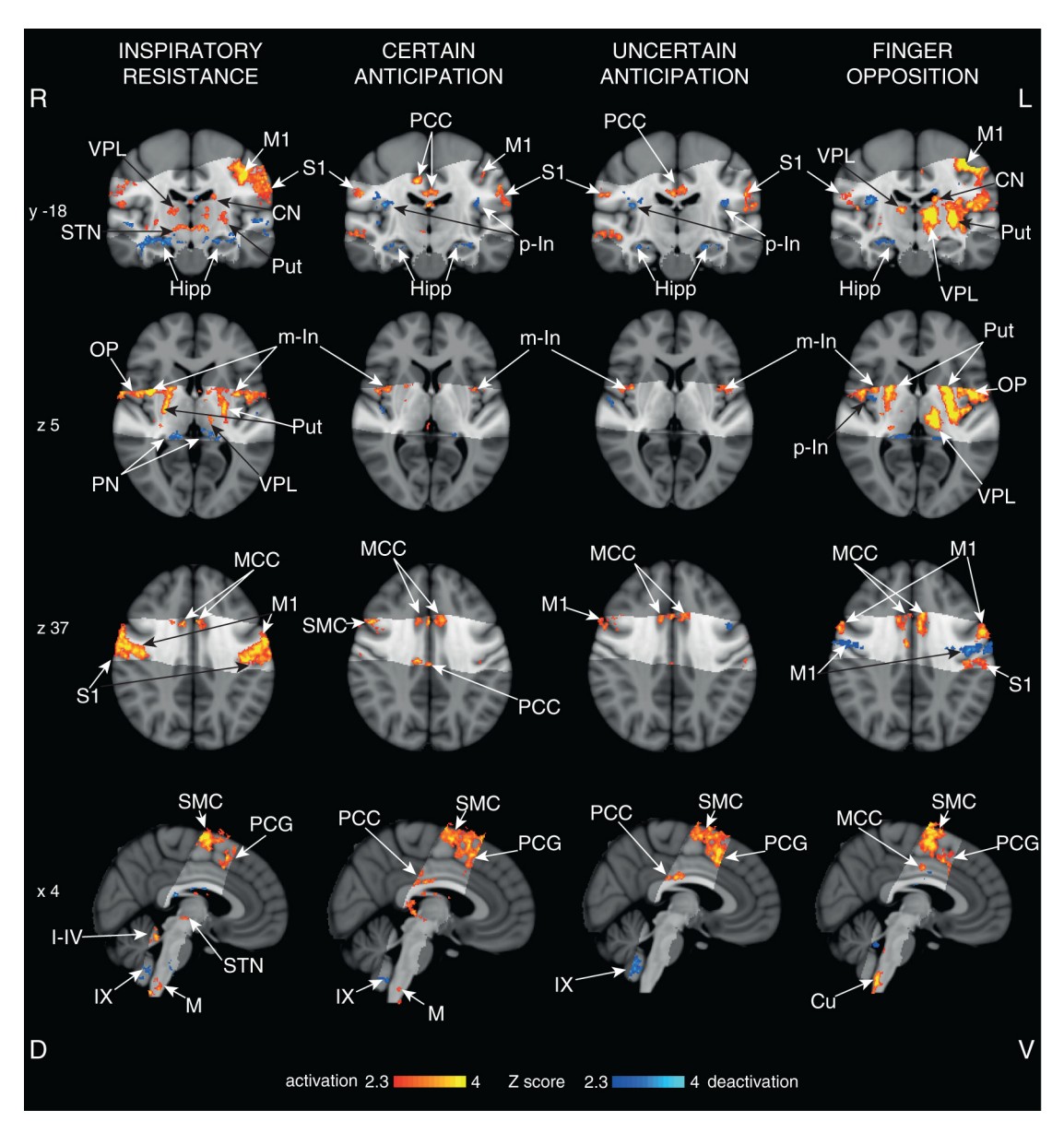

**Figure 5.** Cortical activity with functional tasks. Mean cortical activations and deactivations identified during inspiratory resistance, 100% certain anticipation, 50% uncertain anticipation and finger opposition. The images consist of a colour-rendered statistical map superimposed on a standard (MNI 1 mm³) brain. The bright grey region represents the coverage of the coronal-oblique functional scan. Significant regions are displayed with a threshold $Z>2.3$, with a cluster probability threshold of $p<0.05$ (corrected for multiple comparisons). Abbreviations: VPL, ventral posterior lateral nucleus (thalamus); M1, primary motor cortex; S1, primary sensory cortex; CN, caudate nucleus; Put, putamen; Hipp, hippocampus; STN, subthalamic nucleus; PCC, posterior cingulate cortex; MCC, middle cingulate cortex; p-In, posterior insular; m-In, middle insular; OP, operculum; SMC, supplementary motor cortex; PCG, paracingulate gyrus; PN, posterior nuclei of the thalamus; PAG, periaqueductal gray; M, solitary nucleus of the medulla; Cu, cuneate nucleus (medulla); I-IV, I-IV cerebellar lobe; IX, IX cerebellar lobe. Source files providing peak voxel locations are provided (*Figure 5—source data 1–3*).

The following source data is available for figure 5:

**Source data 1.** Co-ordinates of local maxima of significant increases (activations) and decreases (deactivations) in the BOLD response to inspiratory loading.

**Source data 2.** Co-ordinates of local maxima of significant increases (activations) and decreases (deactivations) in the BOLD response during certain and uncertain anticipation of inspiratory loading.

*Figure 5 continued on next page*

*Figure 5 continued*

**Source data 3.** Co-ordinates of local maxima of significant increases (activations) and decreases (deactivations) in the BOLD response to a finger opposition task.

emotional component of resistance anticipation (*Critchley et al., 2004*; *Gray et al., 2007*). Future work towards understanding whether the role of the lPAG is causative within this anticipatory breathlessness intensity network may be integral to pinpointing perceptual disruptions in chronic sufferers of breathlessness.

### vlPAG in conditioned breathlessness

The increase in BOLD signal identified in the vlPAG implies a role of the vlPAG in the conditioned response to anticipation of breathlessness. Anticipation of resistance also activated a cortical network of motor, sensory and interoceptive areas, indicating the potential position of the vlPAG within a threat detection and passive preparatory network stimulated by a conditioned breathlessness cue. Additionally, although prefrontal cortical areas were not imaged, diffusion tractography has demonstrated that the vlPAG receives the predominant proportion of the input from the prefrontal cortex (*Ezra et al., 2015*), and animal models report direct connections between the posterior orbital frontal and anterior insula cortices to the vlPAG (*An et al., 1998*). Therefore, it is possible that communication between the vlPAG and areas of executive function, interoception and motor preparation are vital to the threat detection and response selection that occur during the cued anticipation of breathlessness, which will be investigated in future work. While this study has made inroads into functionally differentiating the columns of the PAG at high resolution, further research into the intricacies of these communications is needed to fully understand the role of the vlPAG within this network.

Interestingly, there did not appear to be any significant differences (both within the vlPAG and superior cortical network) between uncertain and certain anticipation of resistance, but rather subthreshold vlPAG activity with uncertain anticipation. Furthermore, the reduction in vlPAG activity was paralleled by reduced anxiety and intensity scores in uncertain anticipation, indicating a smaller conditioned response to this cue. This supports the idea that the vlPAG is involved within the threat perception network for breathlessness, and the magnitude of this activity reflects greater conditioning and increased anticipatory preparation. Interestingly, it does not appear that the uncertainty induced in this study drives hypersensitivity and resultant increased anxiety or rating scores (*Table 1*), differing from previous research in pain (*Rhudy and Meagher, 2000*; *Ploghaus et al., 2003*). However,

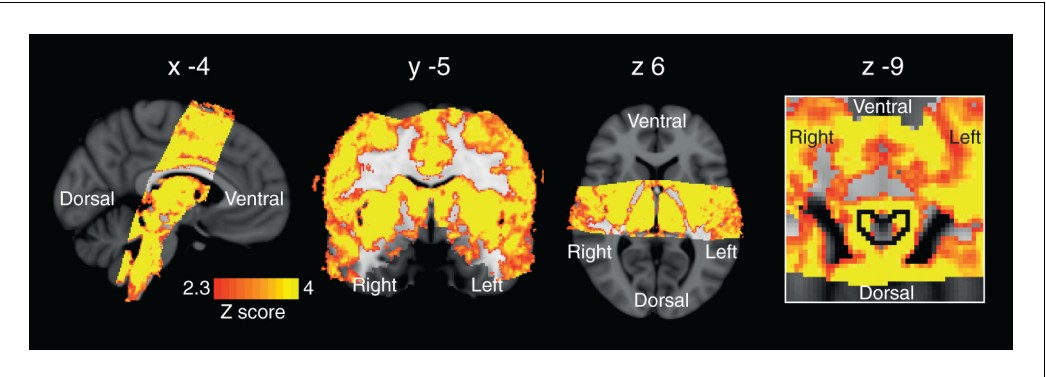

**Figure 6.** Regression of end tidal carbon dioxide effects. Global BOLD signal change correlating with changes in end tidal carbon dioxide ($P_{ET}CO_2$). The image on the right is a zoom to show signal changes within the PAG (outlined in black). Small hypercapnic challenges were administered during rest periods to dissociate hypercapnic effects from respiratory stimuli, and a carbon dioxide ($CO_2$) trace was created by extrapolating between end-tidal $CO_2$ peaks. The images consist of a colour-rendered statistical map superimposed on a standard (MNI 1 mm$^3$) brain. Significant regions are displayed with a threshold $Z>2.3$, with a cluster probability threshold of $p<0.05$ (corrected for multiple comparisons).

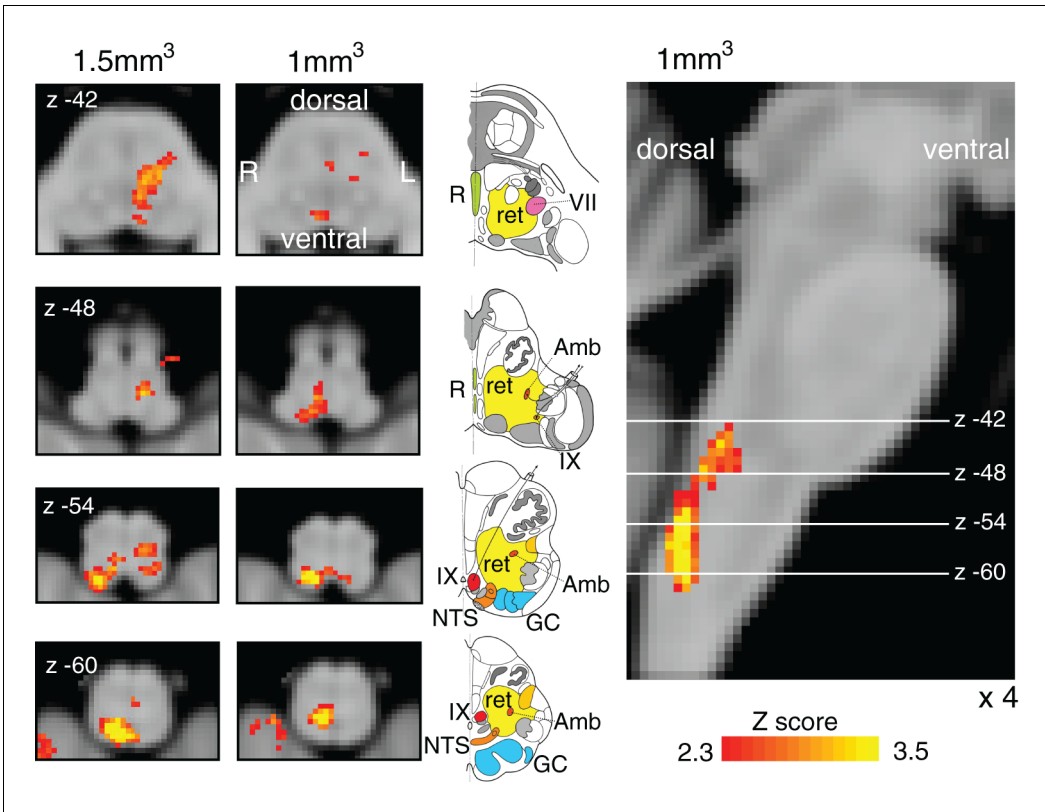

**Figure 7.** Finger opposition functional localiser. Demonstration of the use of finger opposition as a functional localiser in brainstem FMRI in the current study compared to previous results, displaying hypothesised activation in the ipsilateral cuneate nucleus of the medulla (z -54). The 7 T 1 mm$^3$ voxel data is derived from previously-published results (*Faull et al., 2015*) (14 repeats of 15 sec finger opposition, 1 mm$^3$ voxels and TR=5 s), while the 1.5 mm$^3$ voxel data is from the current study (10 repeats of 15 sec finger opposition, 1.5 mm$^3$ voxels and TR=3.11 s). This technique provides confidence in the analysis model and registration accuracy of the current 7 T study. The images consist of a colour-rendered statistical map superimposed on a standard (MNI 1 mm$^3$) brain. Significant regions are displayed with a threshold Z>2.3, with a cluster probability threshold of $p<0.05$ (corrected for multiple comparisons). The sagittal image on the right displays the position of slices, for clarity only displayed from the 7 T 1 mm$^3$ acquisition. Abbreviations: R, raphe nuclei; ret, nuclei reticularis; VII, facial nucleus; Amb, nucleus ambiguous; IX, glossopharyngeal nucleus; NTS, nucleus tractus solitaries; GC, gracile (medial) and cuneate (lateral) nuclei (in blue). R (right) and L (left) indicate image orientation. Original line drawings adapted from *Duvernoy, 1995*.

previous pain research has often used no anticipation cue in conjunction with an unpredictably intense stimulus, while in this study we used an uncertain prediction of a known stimulus. The current methodology allowed us to manipulate the conditioned response to a cue, permitting investigation into the role of the PAG during reduced perception of threat without changing the intensity of the stimulus.

## Cortical respiratory threat network

This study also revealed a cortical and subcortical network of structures that co-activated with PAG columnar activity in these conditions. Anticipation and conscious changes in respiration involve both sensorimotor and affective processing, as adequate ventilation is integral to sustaining life and thus closely monitored by homeostatic mechanisms (*Brannan et al., 2001*; *Dempsey et al., 1985*). Within the limited field of view of this study, the cortical network associated with breathing against an inspiratory load covered a network of primary motor and sensory structures, and the subcortical basal ganglia and insula (*Figure 5*) as well as the lPAG, consistent with previous research using breath holds (*Faull et al., 2015*; *McKay et al., 2008*; *Pattinson et al., 2009*) and hypercapnia-stimulated hyperventilation using PET (*Brannan et al., 2001*). Conversely, during anticipation, vlPAG activity

was paired with less extensive activation of cortical primary motor and sensory structures compared with inspiratory loading, while activity was maintained in preparatory motor structures such as the supplementary motor cortex and basal ganglia (*Groenewegen, 2003*; *Mink, 1996*; *Alexander et al., 1986*). While further research is required to investigate the role of prefrontal brain activity that will be simultaneously occurring within this respiratory threat network during anticipation and inspiratory loading, what will be of great interest is how these distinctly different PAG columnar activations are functionally interacting within this extensive cortical network to influence the perception of respiratory threat during these two conditions.

### Analysis techniques

It is common practice within the learning literature to contrast the conditioned cue that is paired with the stimulus with a cue that is unpaired with the stimulus (*Büchel et al., 1998*; *Gottfried et al., 2002*; *Gottfried and Dolan, 2004*; *LaBar et al., 1998*), which in this case would be the contrast of certain anticipation of resistance with the anticipation of no resistance, respectively. However, this contrast is not feasible beyond targeted PAG column analysis in the current study, as the length of the inspiratory resistance stimulus required to amass statistical power limits the number of possible repeats of each condition. Therefore, beyond the targeted analysis of the vlPAG during certain anticipation of resistance greater than anticipation of no resistance, the anticipation conditions have been analysed against baseline. However, the inclusion of three anticipatory cue conditions does allow greater decorrelation of the general 'cue response' to each anticipation condition in the model. Further studies in this area may look to include more subjects, or fewer anticipation conditions to allow more repeats, enabling contrasts of anticipation of loading against anticipation of no loading in the whole PAG and wider cortex.

The finger opposition task was used as both a control motor task and a methodological validation. Consistent with previous research (*Faull et al., 2015*; *Pattinson et al., 2009*), we hypothesised to see a localised increase in BOLD signal in the ipsilateral cuneate nucleus of the medulla, which is a sensory nucleus in the fine touch and proprioception pathway prior to decussation (*Craven, 2011*). This activation demonstrates the accuracy of registration required to align activations within small brainstem nuclei for group analysis.

Brainstem fMRI is particularly susceptible to low signal to noise when compared to cortical areas. Physiological noise can present a significant problem, due to bulk susceptibility changes with the respiratory cycle, pulsatile movement with the cardiac cycle, and proximity to fluid-filled spaces (*Brookes, et al., 2013*; *Harvey et al., 2008*; *Hayen et al., 2013b*). Special care was taken in this study to address these issues, with ICA denoising used for movement and scanner artefact, and physiological noise modelling and RETROICOR used for slice-wise removal of cardiac and respiratory noise.

### Conclusions

The results of this study suggest that the individual columns of the PAG may be differentially involved in the perception of breathlessness. This study corroborates with recent findings that the lPAG may be involved with the sensorimotor aspect of breathing control during the active response to breathlessness, and top-down anticipatory activity may influence intensity perception of breathlessness. Conversely, the vlPAG appears to be only activated during anticipation of breathlessness, consistent with freezing behaviours reported in animals, with decreased anticipatory cue conditioning resulting in reduced vlPAG activity. We propose that the vlPAG is involved with the learned anticipatory threat detection of a breathlessness stimulus, corroborating with the proposed model of the vlPAG in the passive threat response to an inescapable stressor. In this study we have discriminated differential functional activity within the columns of the PAG in response to threat for the first time in humans, demonstrating the key differential roles of individual columns within the perception of breathlessness.

## Materials and methods

### Subjects

The Oxfordshire Clinical Research Ethics Committee approved the study and volunteers gave written, informed consent. Eighteen healthy, right-handed volunteers (12 males, 6 females; mean age ±

SD, 28 ± 4 years) undertook one training session, followed by one MRI scanning session within 24 hr. Prior to scanning, all subjects were screened for any contraindications to magnetic resonance imaging at 7 Tesla.

## Breathing system

A breathing system was constructed to remotely administer periods of inspiratory resistance during scanning (*Figure 8*). During rest periods, compressed medical air was delivered to the breathing system and gas flow was maintained at a rate that was adequate to allow free breathing, sufficient that the reservoir bag never collapsed on inspiration. During inspiratory resistance, delivery of compressed air was stopped, and once the reservoir bag collapsed, inspiration was through the resistance arm of the circuit inhaling atmospheric air (see *Figure 8* for details).

To minimise the effect of changing arterial oxygen ($O_2$) and carbon dioxide ($CO_2$) levels upon the BOLD signal, the following steps were employed: additional medical oxygen was delivered, and the flow rate was manually adjusted to minimise fluctuations in pressure of end-tidal oxygen ($P_{ET}O_2$), aiming to keep $P_{ET}O_2$ at 18 kPa (very slightly above normal). At designated time points during rest periods of the functional scan, $CO_2$ challenges were administered by switching the flow of compressed air for a 10% $CO_2$ mixture (10% $CO_2$; 21% $O_2$; balance nitrogen) at 20 L/min for periods of 5–10 s, aiming to raise $P_{ET}CO_2$ an equivalent amount as observed during the inspiratory loading periods. The subject's nose was blocked using foam earplugs and they were asked to breathe through their mouth for the duration of the experiment.

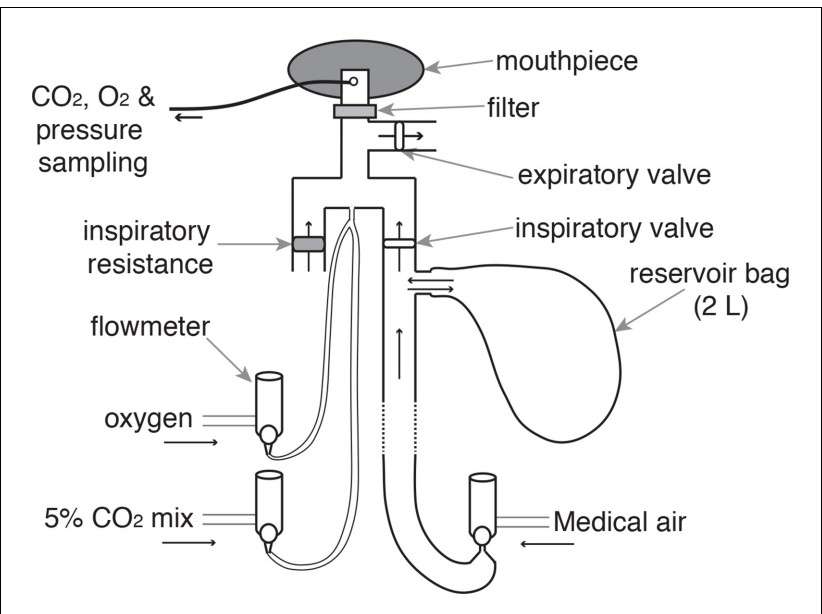

**Figure 8.** Breathing system. Schematic diagram of breathing system that allows remote administrations of inspiratory resistance. Medical air is supplied to the subject, with a reservoir of 2 L. Excess flow and expiration escapes through the one-way expiratory valve, close to the mouth to minimise rebreathing (inspiratory and expiratory valves: Hans Rudolf, Kansas City, MO, USA). Resistive loading is induced by discontinuing the delivery of medical air, forcing the subject to draw air through the resistor (porous glass disc). A diving mouthpiece (Scubapro UK Ltd, Mitcham, UK) connects to a bacterial and viral filter (GVS, Lancashire, UK), sampling lines (Vygon SA, Ecouen, France), connect to a pressure transducer (MP 45, ± 50 cmH$_2$O, Validyne Corp., Northridge, CA, USA) and amplifier (Pressure transducer indicator, PK Morgan Ltd, Kent, UK) for inspiratory pressure readings, and to a gas analyser (Gas Analyser; ADInstruments Ltd, Oxford, United Kingdom) for respiratory gases. A mildly hyperoxic state was achieved through a constant administration of oxygen at a rate of 0.5 L/min. Periodically throughout scanning carbon dioxide challenges were administered to raise $P_{ET}CO_2$ to match the $P_{ET}CO_2$ rise during inspiratory loading periods.

## Stimuli and tasks

The experimental protocol was completed on two occasions; during the conditioning session and repeated in the scanner the following day. The purpose of the conditioning session was for subjects to learn to associate a different symbol (star, triangle or square; randomised order) to three breathing conditions, and the conditioned response to these symbols was then investigated by repeating the protocol with fMRI. The breathing conditions were as follows:

- Certain upcoming inspiratory resistance: symbol presentation always paired with inspiratory resistance
- Uncertain upcoming inspiratory resistance: symbol presentation paired with inspiratory resistance during 50% of the occasions
- No upcoming resistance: symbol presentation is never paired with inspiratory resistance

The certain or uncertain resistance symbol was presented on the screen for 30 sec, which included a 5–15 s anticipation period before the resistance was applied (where applicable). The no resistance symbol was presented for 20 s, and each condition was repeated 10 times in a semi-randomised order (*Figure 9*). A finger opposition task was also included in the protocol, as a brainstem functional localiser for confidence in image registration and analysis techniques (*Faull et al., 2015*; *Pattinson et al., 2009*), where an opposition movement was conducted with the right hand, with the cue 'TAP' presented for 15 s (10 repeats).

Rating scores of breathing difficulty were recorded after every symbol and at the beginning and end of the task, using a visual-analogue scale (VAS) with a sliding bar that the subjects moved between 'Not at all difficult' (0%) and 'Extremely difficult' (100%). Subjects were also asked to rate how anxious each of the symbols made them feel using a VAS between 'Not at all anxious' (0%) and 'Extremely anxious' (100%) immediately following the experimental protocol.

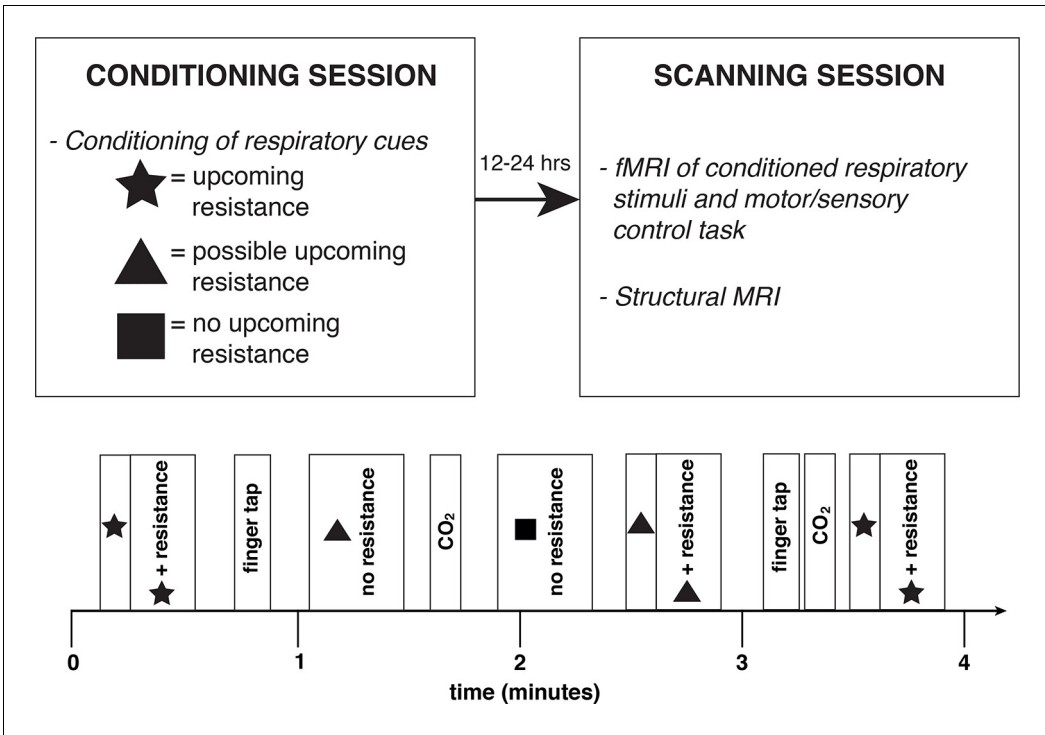

**Figure 9.** Experimental protocol. Study overview (top) and example four minutes of the experimental protocol (bottom), repeated throughout the conditioning and fMRI scanning sessions. Anticipation periods were 5–15 s duration, and resistance periods 15–25 s, and there were 10 repeats of each stimulus.

## Physiological measurements

Using MRI to investigate respiratory control presents methodological challenges that require consideration, particularly in the brainstem (*Brookes, et al., 2013*). We used previously-established methods to decorrelate the effects of hypercapnia from the localised BOLD responses associated with breathing against an inspiratory resistance, using additional, repeated $CO_2$ challenges interspersed during rest periods in the fMRI protocols (*Faull et al., 2015*; *Pattinson et al., 2009*). Additionally, chest movements were measured using respiratory bellows surrounding the chest at the approximate level of the 10th rib, and heart rate was measured using a pulse oximeter (9500 Multigas Monitor, MR Equipment Corp., NY, USA). $P_{ET}CO_2$ and $P_{ET}O_2$ were sampled via a port beside the mouth piece of the breathing system. Expired gases were determined using a rapidly-responding gas analyser (Gas Analyzer; ADInstruments Ltd, Oxford, United Kingdom), and pressure at the mouth was measured using a pressure transducer (MP 45, $\pm$ 50 cmH2O, Validyne Corp., Northridge, CA, USA) connected to an amplifier (Pressure transducer indicator, PK Morgan Ltd, Kent, UK). All physiological measurement devices were connected to a data acquisition device (Powerlab; ADInstruments Ltd, Oxford, United Kingdom) coupled to a desktop computer with recording software (Labchart 7; ADInstruments Ltd, Oxford, United Kingdom).

## Magnetic resonance imaging

MRI was performed with a 7T Siemens Magnetom scanner, with 70 mT/m gradient strength and a 32 channel Rx, single channel birdcage Tx head coil (Nova Medical). The fMRI experimental design is illustrated in *Figure 9*.

### Brainstem BOLD scanning

A T2*-weighted, gradient echo EPI was used for functional scanning. To allow high resolution scanning, a reduced field of view (FOV) was used, with a coronal-oblique slice centered over the brainstem and superior cortical structures. The FOV comprised 36 slices (sequence parameters: TE, 24 ms; TR, 2.11 s; flip angle, 90 deg; voxel size, 1.5 x 1.5 x 1.5 mm; GRAPPA factor, 3; echo spacing, 1 ms; slice acquisition order, posterior-anterior), with 835 volumes (scan duration, 29 mins 40 s).

### Structural scanning

A T1-weighted structural scan (MPRAGE, sequence parameters: TE, 2.96 ms; TR, 2200 ms; flip angle, 7 deg; voxel size, 0.7 x 0.7 x 0.7 mm; inversion time, 1050 ms; bandwidth; 240 Hz/Px; slice orientation, oblique-coronal) was acquired. This scan was used for registration of functional images, and anatomical overlay of brain activations.

### Additional scanning

A single volume whole brain EPI was acquired with 85 slices in the same orientation as the functional scan (matched sequence parameters to the BOLD scan) for registration purposes. Fieldmap scans (sequence parameters: TE1, 4.08 ms; TE2, 5.1 ms; TR, 620 ms; flip angle, 39 deg; voxel size, 2 x 2 x 2 mm) of the $B_0$ field were also acquired in the same orientation to assist distortion-correction of scans.

## Analysis

### Preprocessing

Image preprocessing was performed using the Oxford Centre for Functional Magnetic Resonance Imaging of the Brain Software Library (FMRIB, Oxford, UK; FSL version 5.0.8; http://www.fmrib.ox.ac.uk/fsl/). The following processing methods were used prior to statistical analysis: motion correction (MCFLIRT [*Jenkinson, 2002a*]), removal of the nonbrain structures (skull and surrounding tissue) (BET [*Smith, 2002*]), spatial smoothing using a full-width half-maximum (FWHM) Gaussian kernel of 2 mm, and high-pass temporal filtering (Gaussian-weighted least-squares straight line fitting; 120 s cut-off period) (*Woolrich et al., 2001*). The functional scans were corrected for motion, scanner and cerebro-spinal fluid artefacts using ICA denoising (*Kelly et al., 2010*). Cardiac- and respiratory-related waveforms were used to form voxelwise noise regressors (Physiological noise modelling; FSL version 5.0.8), and the signal associated with these regressors was modelled using retrospective image correction (RETROICOR) (*Brookes, et al., 2013*; *Harvey et al., 2008*). The noise signal

determined by RETROICOR was adjusted for interactions with the ICA denoising to ensure artefactual signal was not reintroduced through the combination of both noise correction techniques.

## Image registration

Careful attention was paid to image registration, as the finer resolution afforded by 7 Tesla MRI requires greater registration accuracy for group statistics to have sufficient power. After preprocessing, the functional scans were registered to the MNI152 (1 mm$^3$) standard space (average T1 brain image constructed from 152 normal subjects at the Montreal Neurological Institute (MNI), Montreal, QC, Canada) using a three-step process.

- Linear registration (FLIRT) with 6 degrees of freedom (DOF) was used to align the partial field of view (FOV) scan to the whole-brain EPI image (*Jenkinson, 2002b*).
- Registration of each subject's whole-brain EPI to T1 structural image was conducted using BBR (Boundary-Based-Registration; part of FEAT: FMRI Expert Analysis Tool, version 6.0) (6 DOF) where (nonlinear) B0 field unwarping was conducted with a combination of FUGUE and BBR tools (*Jenkinson, 2002b*; *Greve and Fischl, 2009*).
- Registration of each subject's T1 structural scan to 1 mm standard space was performed using an affine transformation followed by nonlinear registration (FNIRT) (*Andersson et al., 2009*).

## Voxelwise analysis

Functional data processing was performed using FEAT (FMRI Expert Analysis Tool), part of FSL. The first-level analysis in FEAT incorporated a general linear model (*Woolrich et al., 2004a*), where the finger opposition regressor was derived from the protocol timing values. Inspiratory resistance timings were calculated from the onset to termination of each of the resistance applications from the recorded pressure trace. The anticipation periods were calculated as the time between presentation of the stimulus and onset of inspiratory resistance. Ratings for all respiratory and baseline conditions were included as a rating regressor, that was demeaned against the constant-height inspiratory resistance regressor, to model out variations between the respiratory stimuli. $P_{ET}CO_2$ was included as an additional regressor, de-correlating the $CO_2$ –induced BOLD changes from the respiratory stimuli throughout the functional scan. This trace was formed by linearly interpolating between the expired $CO_2$ peaks. Previous research has indicated that variations in the hemodynamic response function (HRF) are apparent throughout the brainstem and cortex (*Devonshire et al., 2012*; *Handwerker et al., 2004*), and between subjects (*Handwerker et al., 2004*). To account for possible changes in the HRF, including slice-timing delays, we used an optimal basis set of three waveforms (FLOBS: FMRIB's Linear Optimal Basis Sets, default FLOBS supplied in FSL [*Woolrich et al., 2004b*]), instead of the standard gamma waveform. The second and third FLOBS waveforms, which model the temporal and dispersion derivatives, were orthogonalised to the first waveform, of which the parameter estimate was then passed up to the higher level to be used in group analysis. Time-series statistical analysis was performed using FILM, with local autocorrelation correction (*Woolrich et al., 2001*).

Voxelwise statistical analysis was extended to a group level, in a mixed-effects analysis using FLAME (FMRIB's Local Analysis of Mixed Effects) (*Woolrich et al., 2004a*). Z statistic images were thresholded using clusters determined by $Z>2.3$ and a (corrected) cluster significance threshold of $p<0.05$. Univariate analysis of the group mean was performed, and anxiety ratings were used as a covariate of interest in a whole-brain linear regression. Small-volume masks of the vlPAG and lPAG (adapted from diffusion-based segmentation of the human PAG [*Ezra et al., 2015*]) and the whole PAG were used to investigate a-priori areas of interest, using standard cluster thresholding ($Z>2.3$). A further higher-level covariate analysis was performed, which included additional resistance and anxiety scores as demeaned regressors in the higher-level analysis, where the average resistance score across trials was calculated for each subject, and the anxiety score was taken as the anxiety of certain resistance. We included pack years (cigarettes per day x number of years) as a confound regressor to account for any minimal subject history of smoking.

## Acknowledgements

This research was supported by an MRC Centenary Award as part of an MRC Clinician Scientist Fellowship awarded to Kyle TS Pattinson. This research was further supported by the National Institute for Health Research, Oxford Biomedical Research Centre based at Oxford University Hospitals NHS Trust and University of Oxford. Olivia K Faull was supported by the Commonwealth Scholarship Commission. The authors would like to thank Dr Falk Eippert and Dr Vishvarani Wanigasekera for their input and support during the production of this manuscript.

## Additional information

### Funding

| Funder | Grant reference number | Author |
| --- | --- | --- |
| Medical Research Council | Clinician Scientist Fellowship | Kyle T.S Pattinson |
| National Institute for Health Research | | Kyle T.S Pattinson |
| Commonwealth Scholarship Commission | Graduate Student Scholarship | Olivia K Faull |

The funders had no role in study design, data collection and interpretation, or the decision to submit the work for publication.

### Author contributions

OKF, KTSP, Conception and design, Acquisition of data, Analysis and interpretation of data, Drafting or revising the article; MJ, Analysis and interpretation of data, Drafting or revising the article; ME, Conception and design, Analysis and interpretation of data, Drafting or revising the article

### Author ORCIDs

Olivia K Faull, http://orcid.org/0000-0003-0897-7142
Mark Jenkinson, http://orcid.org/0000-0001-6043-0166
Kyle TS Pattinson, http://orcid.org/0000-0003-1353-2199

### Ethics

Human subjects: The Oxfordshire Clinical Research Ethics Committee approved the study and volunteers gave written, informed consent prior to testing.

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
