## [Decision Letter]

Thank you for submitting your work entitled "Conditioned respiratory threat in the subdivisions of the human periaqueductal gray" for consideration by *eLife*. Your article has been favorably evaluated by three peer reviewers: Gert Holstege, Hari Subramanian and Jan-Marino Ramirez, who is a member of our Board of Reviewing Editors. The evaluation was overseen by Eve Marder as the Senior editor.

16) Representation of the coronal schematics of the PAG () is a bit odd when compared to the animal schematics where the dorsal is on the top of the aqueduct and the ventral to the bottom. In their it's the reverse. Any specific reason?

---

## [Author Response]

The reviewers agree with most of the authors' conclusions, but have several minor suggestions to improve this study. It is also important that the authors discuss their conclusions on the role of the PAG in a more general behavioral context, i.e. not only in the context of respiratory control. Minor points:

*1) In the Abstract it is not clear what the volunteers in the scanner did. Please mention.*

*2) In the first paragraph of the main text the authors state that the PAG plays a role in threat identification. The reviewers disagree with this statement. The PAG is used as a tool by the orbitofrontal cortex etc. to excite systems necessary to do so, but it does not identify the threat by itself. Please clarify.*

*3) In the first paragraph of the main text the authors mention descending connections to the respiratory area. Please clarify what these connections are, since they are the basis of this study. By far the most important connection of the PAG regarding respiration control is to the nucleus retroambiguus, the only cell group that has direct access to all the motor neurons innervating muscles involved in thoracic and abdominal pressure control. The first time this was demonstrated was by Holstege, 1989 (JCN 284: 242-252). As you discuss the role of the PAG in the integration of motor and autonomic mechanisms integral to the sensorimotor integration of dyspnea in the PAG, you might want to consider a discussion of the PAG modulation of respiratory neurons in the VLM (Subramanian, 2013; Subramanian and Holstege, 2013). This modulation is dependent on the type of behavior selected. Moreover, the PAG mediation of the hypoxic ventilatory response (Lopes et al., 2014) along with the suffocation alarm system in the PAG (Schmitel et al., 2012) is also an interesting aspect related to dyspnea.*

*4) Please mention that the PAG is not only involved in respiration control, but in almost all basic survival mechanisms. For a review about this see Holstege in Progress in Brain Research 2014 vol. 209 p. 379-406.*

*5) In the second paragraph of the subsection “Periaqueductal gray fMRI analysis”: No significant difference was found between uncertain and certain anticipation of breathlessness. Yet, a difference likely exists, but the statistics are not strong enough.*

*Please mention somewhere that the lateral PAG is involved in fight and flight, and the ventrolateral PAG in freeze. Freeze is based on the fact that movements in the visual field are noticed by the mesencephalon, i.e. by the nucleus of the optic tract etc. which transfers this information to the PAG. Freezing is important for basic survival, and is used by the orbitofrontal cortex to survive dangerous situations.*

*6) In the first paragraph of the subsection “vIPAG in conditioned breathlessness”. Unfortunately, the authors did not image the prefrontal cortical areas. These areas are central in these respiration tasks, even more so than the PAG. Do the authors plan to image these areas in a next study? If so, please mention that.*

*7) In the last paragraph of the subsection “Analysis techniques”. It is surprising that the authors were able to show activations in different parts of the PAG, because the brainstem is continuously moving due to respiratory and blood pressure effects. May be PET-scanning would be a better approach.*

As rightly pointed out, the noise generated by movement in the brainstem/midbrain means that this area is particularly difficult to image non-invasively in humans. While PET scanning is not influenced to quite the same extent as fMRI by this noise, the resolution of PET scanning (> 5 mm[61]) is unfortunately not nearly fine enough to differentiate between PAG columns, nor adjacent anatomical areas such as the cerebral aqueduct. Therefore, we have devoted a considerable amount of time and effort into optimising scanning sequences and analysis techniques to minimise and account for this (and other) noise using fMRI, in conjunction with the ultra high-field 7 Tesla scanner to allow imaging at resolutions of 1-2 mm[61]. This is also exampled in our previous work (Faull et al., 2015 NeuroImage).

*8) In the subsection “Conclusions”: vlPAG, please mention that this structure is involved in freezing behavior.*

*9) Figures. Usually PAG studies show the dorsal PAG on top and the ventral PAG on the bottom of a figure. Could you do the same with your results?*

To clarify the images, we have added image orientations to each of the figures and a key to . We hope this makes the images easier to understand. However, if the reviewers feel strongly that the images should be presented the other way around then we are happy to reconsider this point.

*10) The description of columns in the PAG is based upon its functional specificity along its rostro-caudal axis. In animals we have a clear cytoarchitecture of it [Berman (cat) or Paxinos (rat) atlases], its rostrocaudal dimensions. Along the rostrocaudal extent of the same column, distinct types of autonomic and motor effects can be produced in animals. Also while the dorsolateral and dorsomedial columns are present in the rostral, intermediate and caudal portions of the PAG, the lateral and particularly ventrolateral is present mainly in the intermediate and caudal portions. To be anatomically picky, such a rostrocaudal extension of the PAG has not been cytoarchitecturally examined in the human PAG. Thus it may be better to list them at this point as dorsal, lateral and ventrolateral 'divisions' rather than columns. This is just a suggestion for the authors to think about and present a discussion point as they have used the columnar correlation in their previous Erza et al. as well.*

*11) The PAG's projection connectome with the ACC is not very clear. Most data obtained neuroanatomical tract-tracing in monkeys. Few experiments from Price laboratory identified parallel systems in the rat, although they were largely clustered as orbitofrontal projections and not specifically ACC.*

*12) The authors quote the works of Price and Wager on the anticipation formula. These studies examined pain. The analgesic system is different to that of anticipation of respiratory loading, an autonomic/motor component. Thus exacerbation of result symptoms from a threat/pain point of view may not be the right parallel argument for this study. Moreover, this study does not even require such a comparison. Even if such a comparison is made, there is not any indication about the dorsal PAG in their results, which is involved in analgesia. If the anxiety anticipation has any pain component to it, one would expect the dorsal PAG also to be activated during such interventions? Could the authors make a comment on this?*

*13) It is surprising that no areas of the PAG or cortex significantly scaled with intensity or anxiety ratings during inspiratory loading? What could the reason be? It is also interesting that they did not see BOLD signal increases (for anticipation) in the ACC. Perhaps the authors can discuss this a bit more?*

b) The voxel size used in this study is very small for fMRI (typically >2-3 mm[61] when using < 3 T), while in this study we used 1.5 mm[61] voxels. While the use of the 7 T scanner affords greater intrinsic signal within the voxels, reducing the size decreases this signal within each voxel. Therefore, we have used the 7 T scanner properties to gain greater resolution, rather than the potential increase in signal required to observe scaled activity across subjects with this noisy stimulus.

*14) The authors also show many cortical and sub-cortical structures activated/deactivated during the interventions. It would be useful to link these with PAG function in the Discussion providing some suggestions on possible forebrain-midbrain pathways involved in dyspnea.*

*15) The figure legends can be tightened and made a bit clear.*

16) Representation of the coronal schematics of the PAG () is a bit odd when compared to the animal schematics where the dorsal is on the top of the aqueduct and the ventral to the bottom. In their it's the reverse. Any specific reason?